# Beta-Caryophyllene, a Cannabinoid Receptor Type 2 Selective Agonist, in Emotional and Cognitive Disorders

**DOI:** 10.3390/ijms25063203

**Published:** 2024-03-11

**Authors:** Caterina Ricardi, Serena Barachini, Giorgio Consoli, Donatella Marazziti, Beatrice Polini, Grazia Chiellini

**Affiliations:** 1Department of Pathology, University of Pisa, 56126 Pisa, Italy; c.ricardi@student.unisi.it; 2Hematology Division, Department of Clinical and Experimental Medicine, University of Pisa, 56126 Pisa, Italy; serena.barachini@unipi.it; 3Department of Psychiatry, University of Pisa, 56126 Pisa, Italy; gconsoli@psico.med.unipi.it (G.C.); dmarazzi@med.unipi.it (D.M.); 4UniCamillus, Saint Camillus International University of Health and Medical Sciences, 00151 Rome, Italy

**Keywords:** neuropsychiatric diseases, depression, anxiety, inflammation, neuroinflammation, COVID-19, endocannabinoid system, cannabinoid type 2 receptor, β-caryophyllene

## Abstract

Mental disorders account for one of the most prevalent categories of the burden of disease worldwide, with depression expected to be the largest contributor by 2030, closely followed by anxiety. The COVID-19 pandemic possibly exacerbated these challenges, especially amongst adolescents, who experienced isolation, disrupted routines, and limited healthcare access. Notably, the pandemic has been associated with long-term neurological effects known as “long-COVID”, characterized by both cognitive and psychopathological symptoms. In general, psychiatric disorders, including those related to long-COVID, are supposed to be due to widespread inflammation leading to neuroinflammation. Recently, the endocannabinoid system (ECS) emerged as a potential target for addressing depression and anxiety pathophysiology. Specifically, natural or synthetic cannabinoids, able to selectively interact with cannabinoid type-2 receptor (CB2R), recently revealed new therapeutic potential in neuropsychiatric disorders with limited or absent psychotropic activity. Among the most promising natural CB2R ligands, the bicyclic sesquiterpene β-caryophyllene (BCP) has emerged as an excellent anti-inflammatory and antioxidant therapeutic agent. This review underscores BCP’s immunomodulatory and anti-inflammatory properties, highlighting its therapeutic potential for the management of depression and anxiety.

## 1. Introduction

Mental disorders constitute the predominant portion of the global burden of disease on a worldwide scale. Predictions indicate that by 2030, depression is expected to emerge as the primary contributor to the global disease millstone. Mood-related disorders are identified as the primary contributors, with anxiety-related disorders, substance abuse and schizophrenia following closely [1].

Depression and anxiety disorders are the most prevalent and pervasive psychiatric disorders worldwide. Depression represents a major burden of global disease with a prevalence of about 5%. It is estimated that approximately 25% of the population suffers from a depressive episode during their life span. The major symptoms of depression include persistent depressed or irritable mood, apathy, decreased interest in normal or even enjoyable activities, decreased social contacts, somatic distress and cognitive impairment [2]. Besides the subjective suffering that can be sufficiently severe to induce suicide, such symptoms cause significant disruptions in daily life as well as in familial life, work, study and social adjustment. Not surprisingly, depression is the major cause of global disability and imposes a significant economic burden on the healthcare systems [3].

Although anxiety is a normal reaction that permits us to cope with internal and environmental triggers and challenges, anxiety disorders differ from normal feelings of nervousness or anxiousness and involve excessive fear. Anxiety disorders are prevalent, occurring in about 4% of the global population. The most common anxiety disorders include panic disorder, generalized anxiety disorder, agoraphobia, social anxiety disorder and specific phobias. Individuals grappling with anxiety disorders often contend with persistent alarm state and preoccupations. Additionally, they may experience a broad range of physical manifestations such as perspiration, trembling, dizziness, headache, gastro-intestinal distress and/or a heightened heart rate [2].

The recent Coronavirus disease (COVID-19) pandemic, which caused over two million deaths globally, has raised significant mental health concerns. Noteably, COVID-19 has been increasingly linked to long-term neurological effects, known as “long-COVID”, including cognitive impairment, fatigue, depression and anxiety [4].

The World Health Organization (WHO) reported that the pandemic has primarily affected the mental health of young people and that they are disproportionately at risk of suicidal and self-harming behavior. However, while there is extensive research on the impact on the elderly population, little attention has been given to the psychological toll on adolescent mental health. Adolescents aged 13–17 face unique challenges due to limited psychological resilience, coping abilities and physiological development. Pre-existing mental health conditions may worsen, influenced by factors like isolation, disrupted routines, and limited access to school-based health services. Globally, 10–20% of adolescents experience mental health conditions and the pandemic likely increases their vulnerability [5]. Since the onset of COVID-19, Social Anxiety Disorder (SAD), a long-term anxiety condition characterized by fear of evaluation and avoidance of social situations, is one of the mental illnesses that has increased in young adults [6].

Several psychiatric disorders, including long-COVID syndromes, are supposed to be characterized by a systemic state of sterile inflammation, resulting in the release of Damage-Associated Molecular Patterns (DAMPs), recognized by Pattern Recognition Receptors (PRRs), among which Toll-Like Receptors (TLRs) or NOD-Like Receptors (NLRs) are the most important. Once activated, these receptors induce the release of pro-inflammatory cytokines that generate the inflammation with no involvement of pathogens. This sterile inflammation, often linked to psychiatric disorders, such as episodes of major depression or bipolar disorder, may lead to an increase in blood–brain barrier (BBB) permeability to pro-inflammatory cytokines, DAMPs, or infiltration of leukocytes and macrophages associated with neuroinflammation [7].

In recent years, different studies have suggested the existence of a relationship between alterations in the endocannabinoid system (ECS) and the development of symptoms linked to depression and/or anxiety disorders.

ECS components modulate fear and anxiety-related behavior in both humans and rodents [8,9,10]. Increased ECS signaling is usually followed by reduced conditioned fear and anxiety, whereas the opposite effect is observed when it is inhibited [11,12,13,14]. The involvement of the ECS in modulating psychiatric disorders appears to be due to two G protein-coupled receptors, cannabinoid receptor type-1 (CB1R) and cannabinoid receptor type-2 (CB2R). The key actions of these receptors include the inhibition of adenylate cyclase and voltage-dependent calcium channels, the activation of MAP kinases, and the modulation of potassium channels, resulting in changes in both synaptic function and gene transcription, as well as the migration of neurons and glial cells [15].

CB1R is widely distributed in the central nervous system (CNS), particularly on axon terminals and pre-terminal axon segments, while CB2R is present in microglia and in immune cells and is upregulated during tissue damage or neuroinflammation. CB1R plays a role in motivation and cognition, being involved in GABAergic and glutamatergic pathways. Under physiological conditions, CB2R has been identified in neurons and brain regions extensively associated with emotional, rewarding and cognitive behavior, including the cerebellum, brainstem, amygdala (AMY) and hippocampus (HIPP) [16].

Endogenous ligands (endocannabinoids, ECs) derived from membrane phospholipids are synthesized on demand from arachidonic acid by specific lipase as a response to increased intracellular Ca^2+^ levels and are released in one or two rapid enzymatic steps into the extracellular space without storage in synaptic vesicles. Among ECs, the two most studied ones are anandamide (AEA) and 2-arachidonoylglycerol (2-AG). Anandamide can be metabolized by fatty acid amide hydrolase (FAAH), while three serine hydrolyses, including monoacylglycerol lipase (MAGL), serine hydrolase α-β-hydrolase domain 6 (ABHD6) and serine hydrolase α-β-hydrolase domain 12 (ABH12), account for about 99% of 2-AG hydrolysis in the CNS. Both ECs can undergo metabolism by both cyclooxygenase-2 (COX-2) and N-acylethanolamine acid amidase (NAAA), suggesting that COX-2 inhibitors might elevate endocannabinoid levels [17].

CB2R has been recently identified as a potential immunomodulatory agent in neuroinflammation associated with psychiatric disorders. When activated, CB2R inhibits the release of pro-inflammatory mediators by microglia, resulting in a neuroprotective effect in different neuropsychiatric conditions, such as Alzheimer’s disease (AD), Parkinson’s disease (PD), multiple sclerosis (MS), depression, anxiety and addiction. The molecular mechanisms underlying this anti-inflammatory effect appear to involve the control of microglial activity by CB2R, interfering with the NF-κB pathway and mitogen-activated protein kinase (MAPK) pathways [c-Jun N-terminal Kinase (JNK), ERK, or p38] [18]. Other signaling pathways involved include JAK (Janus kinase)/STAT1 (signal transducer and activator of transcription) or activation of the pCREB-Bcl-2 pathway. It has been suggested that CB2R can shift microglia toward an anti-inflammatory phenotype, with a consequent increase in the release of anti-inflammatory cytokines, and a reduction in inflammatory microglia markers [18]. 

In recent years, the scientific interest in natural compounds as potential novel drugs has grown exponentially, along with the number of trials and studies on nutraceuticals and herbal extracts, which are aimed at examining their effects on various disorders, including neuropsychiatric diseases. The role of cannabimimetic food as a nutraceutical strategy has been proposed for possible therapeutic benefits, with a particular focus on CB2R, whose role in resolving inflammation and pain, relieving stress, boosting immunity and mitigating oxidative stress has been largely demonstrated [19].

Β-caryophyllene (BCP), a natural occurring bicyclic sesquiterpene present in several cannabis and non-cannabis plants, is one of the most studied and promising natural compounds [20]. Sesquiterpenes are some of the prevalent terpenes found in the essential oils of various plants, including *Cannabis sativa*. Terpenes not only convey the smell of the different cannabis flowers, but also have a role in respiration, photosynthesis and defense. Furthermore, beyond their intrinsic therapeutic abilities, a synergy between terpenes and phytocannabinoids has been proposed, suggesting the putative role of this class of compounds for enhancing the beneficial activity of phytocannabinoids on human health [21].

Notably, BCP is widely distributed throughout the plant kingdom, serving as a predominant component in essential oils extracted from over 300 edible plants, including cloves, hops, pepper, oregano, and rosemary. Thanks to its lipophilicity, BCP is highly bioavailable upon oral consumption, and thus, the fruits, florescences, seeds, leaves, oils and plant extracts rich in BCP could be used as useful nutritional or dietary supplements in day-to-day life, promoting health and curbing a large amount of inflammatory diseases [19].

BCP has been identified as a fully selective agonist of CB2R (pKi value = 155 nM), making its putative clinical application free from the psychotropic effects mediated by brain CB1R activation [20]. Although its mechanism of action is not yet fully understood, studies showed that upon binding to the CB2R, BCP can activate several CB2R-mediated intracellular signaling pathways, such as adenylate cyclase inhibition, intracellular calcium release, and mitogen-activated kinases Erk1/2 and p38 activation, representing a valid clinical target for several disorders [19,20].

In addition to CB2R activation, BCP has been recently demonstrated to interact with members of the family of peroxisome proliferator-activated receptors (PPARs), such as PPAR α and γ, which are involved in both metabolic and inflammatory responses [19,20].

The present narrative review aims to provide an overview of the immunomodulatory and anti-inflammatory roles of BCP and comprehensively summarizes the recent insights into the CB2R-mediated pharmacological properties and therapeutic potential of BCP as a novel natural approach for the management of several psychiatric conditions, with a focus on depression and anxiety.

## 2. CB2R as a Potential Modulator of Neuroinflammation and Neuropsychiatric Disorders

Recently, the emergence of CB2R as a potential immunomodulatory agent with specific functions related to cell-type specificity gained considerable interest in scientific and medical research. Given the correlation between inflammatory processes and neuropsychiatric disorders, understanding the role of CB2R in neuroinflammation will possibly help us understand how the modulation of this receptor could be promising for the management of psychiatric diseases.

### 2.1. CB2R in Neuroinflammation

CB2R has been historically recognized as a pivotal regulator of the immune system in the periphery, being highly expressed in a wide range of immune cells. However, it is now acknowledged that CB2R is also expressed within the CNS, specifically in microglia, astrocytes and certain neuronal subpopulations [22].

In particular, microglial CB2R appears to be involved in various conditions and diseases related to neuroinflammation, including AD, PD, MS, depression, anxiety and addiction [23,24,25].

In vitro and in vivo studies suggested that the activation of CB2R could prevent microglial activation and the consequent release of pro-inflammatory mediators (such as IL-1β, TNF-α and IL-6), leading to a neuroprotective effect. Specifically, the modulation of microglia activity is due to the ability of CB2R to interfere with the NF-κB and MAPK pathways, both targets for the regulation of inflammation [26,27].

Furthermore, increasing evidence suggested that modulating inflammation by activating astrocytic CB2R could contribute to supporting neuronal function. In in vitro models of inflamed astrocytes, treatment with CB2R agonists has been found to reduce the release of pro-inflammatory mediators (such as TNF-α, CXCL10, CCL2 and CCL5) [28], and to inhibit iNOS, COX-2 protein and TNF-α and IL-1β mRNA upregulation [29]. Interestingly, it has been reported that astrocyte CB2R signaling is related to the inhibition of p38 phosphorylation, which is associated with anti-inflammatory effects [30,31]. Moreover, in vitro studies showed how the activation of CB2R at the level of astrocytes alleviated neuroinflammation and protected BBB from increased permeability [26,32]. This particular action appeared to be primarily mediated by a reduced expression of an astrocytic enzyme involved in BBB leakage, the TLR4/Matrix metallopeptidase 9 (MMP-9), and MyD88/NFkB level pathway [32].

Although the most studied mechanisms of CB2R-mediated neuroprotection are those involving microglia and astrocytes, this protective effect could also be attributed to the activation of CB2R in neurons, while it is still unclear whether immune functions could be modulated by neuronal CB2R and what mechanism is actually involved. In particular, studies revealed that treatment with CB2R agonists was able to induce neurogenesis and an anti-inflammatory cytokine profile [33] and to promote neuronal proliferation [34].

Below, we briefly review the evidence supporting the putative role of CB2R in various neuropsychiatric conditions, highlighting its involvement in the regulation of physiological functions.

### 2.2. CB2R in Neuropsychiatric Disorders

Expanding on the current state of neuropsychiatric disorder management, it is evident that the challenges extend beyond the absence of effective treatment. The pervasive issue of adverse drug effects associated with conventional treatments adds another layer of complexity [35]. This underscores the urgency of exploring novel therapeutic targets for the development of safe and effective medications capable not only of alleviating symptoms, but also of hampering the progression of these debilitating and common disorders. For this reason, it is necessary to characterize the underlying mechanisms of the development and progression of neuropsychiatric disorders in more detail, with the final goal of identifying new clinical targets. In this respect, cumulative evidence suggested CB2R as a promising therapeutic option for the treatment of emotional and cognitive disorders. The focus on CB2R has emerged as a compelling area of study, with multiple lines of research pointing towards its potential as an effective therapeutic target for various neuropsychiatric conditions. In particular, a large body of evidence strongly suggests that CB2R activation can produce antidepressant and anxiolytic effects, making this receptor a useful target for the pharmacological treatment of depression and anxiety disorders [18]. 

#### 2.2.1. CB2R in Depression

As previously mentioned, depression is a common but serious mood disorder. It causes severe symptoms that affect how a person feels, thinks, and handles daily activities, such as sleeping, eating or working. Current treatment options are often not sufficiently effective, with issues such as delayed onset of biological action and a high potential to cause adverse effects. Recent pharmacological and genetic findings strongly support the role of CB2R in the regulation of depressive disorders and point out this receptor as a new potential key target in the treatment of different mood-related disorders. The first evidence of the putative importance of CB2R for the treatment of psychiatric disorders came from the detection of lower CB2R expression in different animal models of depression, in particular at the level of HIPP, AMY and the frontal cortex, critical brain regions for emotional reactivity and decision-making [36,37]. A study found a reduction in CB2R expression at the level of AMY and prefrontal cortex in the postmortem brain of suicide victims [38]. Besides, Q63R polymorphism of CB2R, which could alter the receptor function and reduce the physiological responsiveness to CB2R ligands, has been found to be highly incident in depressive patients [39]. The role of CB2R in emotional disorders has been further supported by pharmacological studies performed in rodents, demonstrating that, on the one hand, the overexpression of CB2R in neurons and glia is related to a decreased depressive-like behavior, and on the other, the deletion of CB2R is associated with an increased susceptibility to these responses [40]. Similarly, pharmacological studies explored the effects of CB2R agonists in mouse models of depression, revealing a reduction in depressive behaviors, further providing valuable insights into the potential therapeutic application of CB2R activation for the treatment of neuropsychiatric diseases. With respect to Δ^9^-tetrahydrocannabinol (THC) and cannabidiol (CBD), two of the main phytocannabinoids, studies both in human and in rodents demonstrated their antidepressant-like effects. In rats, administration of THC or CBD has been shown to produce antidepressant-like effects. In humans, co-administration of THC and CBD has been observed to reduce symptoms of depression [21].

In the context of the neuroinflammation hypothesis of depression [41,42,43], lipopolysaccharide (LPS) is commonly used to induce depressive-like behaviors and cognitive impairment [44,45,46]. Interestingly this inflammatory model has been found to induce a significant upregulation of CB2R at the microglia level [47]. Several pharmacological studies with CB2R agonists showed improvements in LPS-induced behavioral alteration and neuroinflammation and reductions in the LPS-induced release of pro-inflammatory cytokines at the brain level [48,49,50]. Earlier research highlighted the modulatory role of CB2R in mouse stress-induced excitotoxicity and neuroinflammation, demonstrating that activation of CB2R was able to prevent oxidative stress and the release of pro-inflammatory cytokines [51].

Neuroplasticity is also influenced by CB2R modulation. In this respect, clinical and animal studies have highlighted the link between low levels of brain-derived neurotrophic factor (BDNF) and the development of behavioral symptoms of depression. BDNF is one of the main neurotrophic factors involved in neurogenesis and plays an important role in modulating neuroplasticity. It has been considered a biomarker for depression for decades [52], although controversies do exist on this matter, while being implicated in the development of both depressive symptoms and antidepressant response [53]. The overexpression of CB2R was able to increase BDNF levels in the HIPP, while depletion of CB2R was related to low BDNF [54]. Treatment with CB2R agonists significantly increased CB2R and BDNF gene expression in the HIPP of stress-induced mice, and CB2R activation was found to significantly upregulate BDNF while reducing several neuroinflammatory markers [55,56].

#### 2.2.2. CB2R in Anxiety and Anxiety-Related Disorders

Investigations of the connection between CB2R and anxiety continue to be an active field of research, with current findings lacking conclusive outcomes. Nevertheless, there are indications proposing that stimulating CB2R could potentially yield to anti-anxiety effects, primarily by influencing immune responses and dampening inflammation.

Recent studies have placed a heightened emphasis on exploring the impact of CB2R on anxiety, particularly in the context of rodents exposed to various stress-inducing scenarios. In rats, when maternal deprivation (MD) was used as a model to assess the long-term consequences of early life stress, a significantly increased expression of CB2R was observed in HIPP [57]. In mice exposed to social defeat, an augmented expression of CB2R in the HIPP was also observed [58]. Notably, studies revealed that mice overexpressing CB2R showed a resistance to anxiogenic-like stimuli, whereas mice lacking CB2R exhibited an anxious attitude [40,59]. 

The crosstalk between CB2R and anxiety-like behavior has also been highlighted by pharmacological studies. Administration of CB2R agonists such as β-caryophyllene (BCP) and JWH133 was observed to produce anxiolytic effects in rodents [60,61]. Interestingly, in mice chronic treatment with AM630, a CB2R antagonist, was observed to produce a significant increase of CB2R expression, in turn leading to antidepressant and anti-anxiogenic effects [56]. Studies in healthy volunteers demonstrated that THC and CBD can exert opposing effects on anxiety. THC itself has anxiogenic activity, but this effect is diminished when THC is administered in combination with CBD. In contrast, CBD administered alone has anxiolytic properties [62].

Anxiety-related disorders include SAD, phobias, panic disorder, and separation anxiety. Post-traumatic stress disorder (PTSD) and obsessive compulsive disorder (OCD), although closely linked to anxiety, have a distinct etiology and symptomatology that may justify a separate classification within the broad spectrum of mental health disorders. Preclinical studies on anxiety-related disorders in rodent models, characterized by abnormally persistent fear memories, such as PTSD and phobias, showed that CBD treatment could provide relief from symptoms [62]. Thus, the use of CBD and other medicinal cannabinoids has drawn increasing interest as a possible treatment strategy for the improvement of overall PTSD symptomatology as well as specific symptom domains, including sleep disorders, arousal disturbances and suicidal thoughts, also influencing quality of life, pain and social impact [63].

## 3. BCP as a Functional CB2R Agonist to Target Neuropsychiatric Disorders

Several studies demonstrated the pharmacological effects and therapeutic potential of BCP, such as cardioprotective, gastroprotective, neuroprotective, hepatoprotective, nephroprotective, chemopreventive, antioxidant, anti-inflammatory, analgesic, and immunomodulatory properties [20]. In particular, BCP has emerged as a potent anti-inflammatory and immunomodulatory agent (Figure 1), highlighting its emerging role as a candidate for the treatment of several pathological conditions [19].

BCP binds to CB2R, thereby modulating critical cellular signaling pathways associated with oxidative stress, apoptosis, insulin resistance and neuroinflammation. BCP regulates the expression of ROS, SOD and apoptosis-related proteins (such as BAX, Bcl-2 and p53) through the PI3K/AKT pathway, effectively reducing oxidative stress and apoptosis. By influencing the PGC-1α and AMPK/CREB signaling pathways, BCP enhances hippocampal BDNF levels while reducing hippocampal COX-2 expression, ultimately promoting neuroprotection. Additionally, BCP modulates the SIRT1/PGC-1α and PPARγ pathways, leading to decreased LDL and triglyceride levels and increased HDL levels, resulting in improved plasma insulin levels. This modulation also reduces the release of inflammatory agents, contributing to a decrease in overall inflammation.

### 3.1. Immunomodulatory Roles of BCP

CB2R are crucial targets for immune regulation, being primarily located in immune cells [64]. A large variety of studies highlighted the therapeutic advantages of BCP in suppressing the immuno-inflammatory cascade upon CB2R activation. In infants infected with acute respiratory syncytial virus, CB2R activation by BCP was reported to reduce lung pathology by lowering levels of cytokines and chemokines [65]. In patients living with HIV, CB2R activation was found to impede productive infection and viral transmission through a crosstalk/interaction with CB2R [66], and by inhibiting virus replication in monocytes and macrophages [67]. Recent findings demonstrated that BCP modulates systemic and local immunity in an experimental autoimmune encephalomyelitis model [68]. The immunomodulatory effects were attributed to the ability of BCP to inhibit CD4+ and CD8+ T lymphocytes and proinflammatory cytokines [69]. The immunomodulatory activity of BCP was further explained by an enhanced phagocytic capability and macrophages activation [70]. Moreover, BCP exerted a robust immunomodulatory effect by inhibiting both pro-inflammatory cytokines in primary murine splenocytes [71].

In a recent study by Brito et al. [72], BCP demonstrated its ability to alleviate lung tissue damage in a murine model of sepsis by inhibiting oxidative stress, the release of inflammatory mediators, leucocyte recruitment, and bacteremia. 

BCP has been also reported to alleviate leukocyte adhesion to the endothelium, reduce oxidative stress, suppress systemic inflammatory mediators, improve microcirculatory function, decrease bacteremia and alleviate lung injury [73,74]. 

The protective role of BCP against neurological deficits and neuroinflammation has been documented in experimental models, such as middle cerebral artery occlusion-induced cerebral ischemia. Its protective effects, via CB2R activation, include the suppression of oxidative stress, inflammatory mediators, apoptosis and reduction in brain edema, as well as the preservation of tight junction proteins and repair of the blood–brain barrier [75,76]. The molecular mechanisms involved include the downregulation of the TLR4 pathways to suppress inflammation and polarizing microglial phenotype from pro-inflammatory to anti-inflammatory [76], the activation of PI3K/Akt signaling pathway to suppress apoptosis, the upregulation of AMPK/CREB signaling [77] and the upregulation of the Nrf2/HO-1 pathway to suppress oxidative stress and apoptosis [78].

Additionally, BCP has been observed to attenuate neuronal necrosis and the expression of receptor-interaction protein kinase-1 (RIPK1), receptor-interaction protein kinase-3 (RIPK3) and mixed-lineage kinase domain-like protein (MLKL) in cerebral ischemia. This attenuation occurs through the inhibition of high-mobility group box 1 (HMGB1)-Toll-like receptor 4 (TLR4) signaling pathways and proinflammatory cytokines [79].

### 3.2. Anti-Inflammatory Roles of BCP

The primary observed pharmacological and molecular mechanism of BCP involves the inhibition of pro-inflammatory cytokines, NF-κB, adhesion molecules, and chemokines. This is followed by the modulation of signaling pathways, primarily including Toll-like receptors, opioid receptors, SIRT1/PGC-1α, AMPK/CREB, MAPK/ERK, Nrf2/Keap1/HO-1 and the activation of nuclear PPARs. These receptors play a pivotal role in controlling lipid and glucose homeostasis, as well as inflammatory responses [80]. The activation of PPARs through BCP may effectively delay the onset of the cytokine storm originating from resident macrophages.

BCP suppresses systemic inflammation in different organs, including the brain, by inhibiting the release of proinflammatory cytokines and other inflammatory mediators, as well as related signaling pathways [81]. Gertsch and colleagues [82] demonstrated that BCP inhibits lipopolysaccharide/endotoxin (LPS)-induced phosphorylation of kinases ERK1/2 and JNK1/2 in macrophages.

Additionally BCP exhibits anti-inflammatory effects by mediating histaminergic and arachidonic acid pathways [83].

The CB2R-dependent anti-inflammatory mechanism of BCP has been demonstrated in various conditions and diseases, including neuropathic pain [84,85], bipolar disorders [86], autoimmune encephalomyelitis/multiple sclerosis [69,87], neurocognitive disorders [88,89], metabolic and neurobehavioral alterations [90], peripheral neuropathy [91], vascular dementia [92], dopaminergic neurodegeneration/PD [93], AD [94] and cerebral ischemia–reperfusion [95].

In PD, the progressive loss of dopaminergic neurons in the substantia nigra pars compacta leads to symptoms like tremors and bradykinesia. Mitochondrial dysfunction, free radical accumulation, and apoptosis contribute to PD pathogenesis. Activation of CB2R demonstrated neuroprotective effects by suppressing neuroinflammation, restoring neuronal function, and promoting neuronal survival. BCP showed promising results in mitigate PD-like pathology induced by neurotoxic agents by specifically activating CB2R, as observed in both animal models [96,97] and human neuroblastoma cells [98] subjected to PD-related neurotoxicity.

Alzheimer’s disease (AD) is characterized by cognitive decline and neurodegeneration, including defective cholinergic neurotransmission and the presence of neurofibrillary tangles, amyloid plaques and neuroinflammation. CB2R were found to be expressed near amyloid plaques [99] and their activation has been linked to improved cognition and reduced neuroinflammation in AD animal models. BCP demonstrated cognitive protection and reduced amyloid deposition in a large variety of in vitro and in vivo models [19].

Vascular dementia (VD), stemming from recurrent ischemic cerebrovascular events, ranks as the second most prevalent cause of dementia after AD. CB2R activation demonstrates anti-ischemic effects by reducing leukocyte infiltration, vascular adhesion, and proinflammatory cytokine release, thereby alleviating neuroinflammation [100,101]. In a rat VD model [92], BCP complex with hydroxypropyl-β-cyclodextrin (HPbCD/BCP) was revealed to effectively improve cognitive function and mitigate memory deficits, as well as to reduce abnormal hippocampal neurons.

### 3.3. Beta-Caryophyllene May Be Prospective in the Treatment of Anxiety and Depression

Significant findings suggested that BCP may hold clinical value for treating depression and anxiety disorders by addressing both behavioral and inflammatory responses to chronic stress. In Table 1, a summary of the modulatory effects of BCP detected in in vitro and in vivo models of depression and anxiety is reported.

Interestingly, in 2014, Bahi et al. analyzed BCP’s anxiolytic and antidepressant effects in a mouse model of anxiety and depression. After administration of BCP (50 mg/kg, i.p.) in C57BL/6 mice, an improvement in sociability and reduction in anxiety in the elevated plus maze test (EPM), open field test (OFT) and marble burying test (MBT) were reported. Moreover, BCP administration produced an antidepressant-like activity when assessed in behavioral test for depression-like behavior, including the tail suspension test (TST), forced swim test (FST) and novelty-suppressed feeding (NSF) test. These anxiolytic- and antidepressant-like effects of BCP were reversed after pretreatment with the CB2R-selective antagonist AM630, clearly demonstrating CB2R-mediated activity [60].

In a different study, Hwang et al. [102] investigated the antidepressant-like effect of BCP on rat chronic restraint plus stress (CR^+^S)-induced depression. Chronic administration of BCP (25, 50 and 100 mg/kg, i.p.) resulted in a significant decrease in behavioral impairment, as detected from reduced immobility in TST and FST behavioral tests. Furthermore, hippocampal expression of neurotrophic, inflammatory and CB2R levels were measured, revealing a stress-related increase in COX-2 and a decrease in BDNF and CB2R. Hippocampal alterations were ameliorated by BCP, suggesting its neurotrophic and anti-inflammatory action at brain level. In addition, electrophysiological measurements in LPS- treated organotypic hippocampal slices were assessed to evaluate the effects of BCP on inflammatory processes and depression. It is well known that LPS- associated neuroinflammation reduces hippocampal BDNF expression, exerting depression-like behavioral changes, and intensifies hippocampal long-term depression (LTD). LTD is a form of synaptic plasticity characterized by a long-lasting decrease in synaptic strength, which is associated with several physiological functions such as learning, memory and fear conditioning. In this study, BCP was able to reverse LPS-induced augmentation of LTD, which is a physiological factor likely associated with antidepressant activity. These results support BCP’s potential in alleviating the symptoms of depression in relation to the suppression of inflammatory response [102].

Interestingly, in a study performed to assess the effects of BCP on high-fat/fructose diet (HFFD)-induced metabolic and neurobehavioral changes in Wistar rats, BCP (30 mg/kg/day, p.o., 4 weeks) was found to alleviate insulin resistance, oxidative stress, neuroinflammation, depressive-like behaviors and memory deficits. In particular, a reduction in depression and anxiety-like behaviors after treatment with BCP was detected with EPM, OFT, light/dark box (LDB), two-trial Y-maze and FST tests. CB2R, together with PPARγ, was demonstrated to be involved in the anti-inflammatory, anxiolytic and antioxidant effects of BCP [90].

The anxiolytic- and antidepressant-like effects of BCP were supported by further data.

A study evaluated the neurobehavioral effects of BCP in Swiss mice, showing that BCP (200 mg/kg) was able to re-establish the anxiety parameters in the EPM and NSF tests [103].

In Swiss albino mice the anxiolytic effects of BCP was tested by assessing the EPM, rotarod, and LDB tests, which suggested that BCP (10, 25, 50 mg/kg, i.p.) exerted an anxiolytic-like effect in a dose-dependent manner [104].

In another study on Swiss mice, the behavioral effects of oral treatment (50, 100 and 200 mg/kg, p.o.) with the essential oil (EO) derived from *S. odoratissima* leaves and its main component, BCP, was examined. The rota-road, OFT, pentobarbital-induced sleep, hole-board, EPM and LDB tests suggested an anxiolytic-like effect of BCP [105].

Terpenes have been suggested to have a positive impact on some psychiatric endophenotypes [106]. For instance, propolis essential oil (PEO) that contains several terpenes, including BCP, revealed a significant improvement in anxiety-like behavior in restraint-stressed ICR mice. In particular, PEO (50, 100 and 200 mg/kg, p.o., 14 days) led to a significant mitigation of anxiety, as emerged from the elevated plus maze (EPM) test [107]. The presence of terpenes has been proposed to enhance the beneficial activity of phytocannabinoids in humans [21], and therefore, the potential therapeutic value of adding cannabis-derived compounds such as BCP to the treatment with CBD by itself or in coadministration with THC in patients suffering from emotional disorders has been suggested.

Viphyllin^TM^, a standardized extract from black pepper (*Piper nigrum*) seeds containing 30% BCP, was demonstrated to mitigate intestinal inflammation, oxidative stress, and anxiety-like behavior in dextran sodium sulfate (DSS)-induced colitis mice, when orally administrated at 50 and 100 mg/kg doses [108].

When used in zebrafish behavioral assays, including an open field-exploration test and a novel object approach test, BCP revealed no effect on anxiety and swimming velocity, while decreasing immobility (Table 1). BCP had only a sedative effect at the highest dose tested [109]. Taken together, the results of this study indicated a potential dose-dependent downward trend in anxiety levels, which may require an increased dosage. However, the poor solubility in water, associated with high volatility, sensitivity to light and the oxidizing action of oxygen, can contribute to reducing BCP bioavailability, suggesting that a different administration model should be used. 

In a model of streptozotocin (STZ)-induced experimental diabetic mice (BALB/c), the ability of BCP to attenuate depression was assessed. Depression-like behavior was evaluated with TST, and chronic administration of BCP (10 mg/kg, p.o.) showed a significant decrease of immobility, suggesting that depressive-like behavior could be prevented or reduced by this CB2R natural agonist. Moreover, chronic administration of BCP led to decreased levels of IL-1β, TNF-α, and IL-6 [110].

In conclusion, numerous animal studies have demonstrated that BCP has the potential to ameliorate behavioral and inflammatory responses related to chronic and acute stress, suggesting its potential clinical value for the treatment of depression and anxiety. However, it is important to underline that the application of this natural compound in human therapy still requires extensive investigation. To date, no clinical trials have been conducted to investigate the CB2R-dependent protective mechanisms of BCP on anxiety and depression.ijms-25-03203-t001_Table 1Table 1Summary of the modulatory effects of BCP in in vitro and in vivo models of depression and anxiety. Upward (↑) and downward (↓) arrows indicate increase and decrease of the corresponding effect of BCP, respectively.TypeStrainParadigmBehavioral EffectsNeurochemical EffectsReferencesBCP(acute; 50 mg/kg; i.p.)C57BL/6 miceacute stress↑ sociability↓ anxiety↓ depressive-like behavior
[60]BCP(chronic; 25, 50, 100 mg/kg; i.p.)Sprague Dawley ratsCR+S↓ depressive-like behaviorin the HIPP:↓ COX-2↑ CB2R↑ BDNF[102]Organotypic hippocampal slicesLPS (1 µg/mL)
↓ LTDBCP(chronic; 30 mg/kg; p.o.)Wistar ratsHFFD↓ Anxiety↓ Depressive-like behavior↓ Memory deficits↓ Insulin resistance↓ Oxidative stress↓ Neuroinflammation[90]BCP(200 mg/kg)Swiss miceAcute stress↓ Anxiety
[103]BCP(acute; 10, 25, 50 mg/kg; i.p.)Swiss miceAcute stress↓ Anxiety
[104]BCP(acute; 50, 100, 200 mg/kg; p.o.)Swiss miceAcute stress↓ Anxiety
[105]PEO(chronic; 50, 100, 200 mg/kg; p.o.)ICR miceRestraint stress↓ Anxiety
[107]Viphyllin^TM^
(chronic; 50, 100 mg/kg, p.o.)MiceDSS-induced anxiety↓ Anxiety
[108]BCP(chronic; 0.02, 0.2, 2, 4%; p.o. in dosing beaker)ZebrafishAcute stress↓ Immobility at the highest dose used (i.e., 4%)-No effect on anxiety

[109]BCP(chronic; 10 mg/kg; p.o.)BALB/c miceSTZ-induced diabetes↓ Depressive-like behavior↓ IL-1β↓ TNFα↓ IL-6[110]

### 3.4. BCP May Be Prospective in COVID-19-Associated Psychiatric Disorders

The interplay between COVID-19 and neurological dysfunction is caused by an enhanced peripheral inflammation characterized by the release of cytokines and chemokines into the bloodstream and the modulation of immune cells. These peripheral processes may lead to dysfunction in the BBB, characterized by compromised vessels that allow the infiltration of inflammatory molecules and cells into the CNS. Astrocytes and microglia respond to these inflammatory stimuli, increasing the immune response by further releasing cytokines, chemokines, and reactive oxygen species, collectively contributing to the potential loss of neurons and/or synapses. All these events may result in cognitive impairment and various other forms of neurological dysfunction [111].

Many phytocannabinoids and terpenes have been shown to modulate the trinity of infection, inflammation and immunity in COVID-19 [112,113,114,115,116,117,118], suggesting their valuable approach to disrupt this vicious cycle of increasing neuroinflammation [119]. Among them, BCP attracted a great deal of attention. As a full CB2R agonist, BCP exhibits significant therapeutic potential due to its anti-inflammatory and immunomodulatory properties with no psychotropic effects, potentially mitigating the severity and progression of COVID-19 by modulating infection, immunity and inflammation. The anti-inflammatory activity of BCP, affecting multiple pathways and mediators, including the inhibition of pro-inflammatory cytokines, chemokines and adhesion molecules, holds promise as a pharmacological and nutritional strategy to counteract the cytokine storm associated with COVID-19 mortality. The ability of BCP to enhance host cellular immunity, coupled with its antiviral, antibacterial and antioxidant effects, may contribute to symptom control, prevent secondary infections and complications, and inhibit disease progression. Additionally BCP shows potential in protecting against risk factors, impeding virus entry, and mitigating organ damage caused by SARS-CoV-2 across different organ systems [112]. 

### 3.5. Safety and Toxicity of BCP

Recently, BCP has been included in the list of compounds generally recognized as safe by the United States Food and Drug Administration (USFDA), which approved its use as an additive and preservative in food products and beverages. [112].

As the majority of the available studies demonstrated, BCP exerts negligible toxicity and tissue-protective effects. A chemopreventive effect and the absence of adverse effects, genotoxicity, mutagenicity and clastogenicity have been largely demonstrated in different experimental models [120,121,122]. In a 2019 study, BCP was shown to modulate the expression of drug-metabolizing enzymes, which may influence the bioavailability and efficacy of concomitantly administered drugs [123]. Potential synergic and/or additive actions of BCP with many drugs (i.e., azithromycin, atovaquone, metaxolone, imipramine, fluoxetine, docosahexaenoic acid, curcumine, baicalein and catechin) and vitamins have been recently proposed [112]. Furthermore, BCP has been discovered to enhance the therapeutic efficacy of immunosuppressive drugs while concurrently diminishing their side effects. This finding convincingly demonstrates BCP’s ability to mitigate organ injuries induced by drugs or xenobiotics [124].

Even though comprehensive clinical studies might be scarce or limited, significant results that reinforced the potential benefit of BCP on human health were obtained in recent years.

For example, in patients with peptic ulcers, BCP administration was demonstrated to improve dyspepsia symptoms, nausea and epigastric pain and to mediate the inhibition of proinflammatory cytokines [125]. In patients with diabetic painful distal symmetric polyneuropathy, one of the most common and invalidating complications of diabetes mellitus, BCP showed a significant reduction in pain with good tolerance and no adverse effects [126]. In a recent placebo-controlled clinical study in patients with hand arthritis, topically applied BCP was found to be safe, well tolerated and beneficial in reducing pain and inflammation [127]. In 2020, a clinical study on a sample of nineteen women reported that administration of BCP by inhalation was able to improve the libido by significantly increasing the salivary concentration of testosterone without affecting the estrogen concentration and in the total absence of toxic effects [128].

### 3.6. Dosage Forms and Pharmaceutical Development of BCP

It is widely known that BCP exhibits high lipophilicity, low water solubility and susceptibility to oxidation upon air exposure. To address its limited bioavailability, numerous innovative drug delivery systems have been developed. A large variety of formulations, including liposomes, nanoemulsions, nanofibers, microemulsions, nanoparticles, micelles, phospholipid complexes, nanocarriers, nanocomposites, hydrogels and matrix formulations utilizing cyclodextrin, have been devised to augment the solubility, stability and release kinetics of BCP [129]. 

A 2021 study successfully led to the development of a formulation containing BCP nanoparticles that significantly improved BCP bioavailability, promoting transport across the BBB and CNS delivery. This represents the completion of the first stage of development and optimization of a therapeutic prototype that could be applied towards neurodegeneration and to any condition that could benefit from cannabinoid non-psychoactive pharmacological pathways, particularly immunomodulatory and anti-inflammatory pathways. Furthermore, the newly developed formulation is suitable for nasal administration, providing the advantages of not being an invasive method, improving patient compliance and enhancing absorption with direct access to the CNS and the bloodstream [130].

An additional study in 2022 successfully proposed a novel self-emulsifying drug delivery system (SEDDS) to enhance the oral bioavailability of BCP [131]. The objective of the study was to compare the oral bioavailability of SEDDS-formulated BCP (BCP-SEDDS), based on the VESIsorb^®^ formulation technology [132], versus administration of BCP neat oil in 24 healthy subjects under fasting conditions. Overall, when BCP was administered as BCP-SEDDS, the oral bioavailability was significantly higher and BCP was absorbed significantly faster compared to when it was administered as BCP neat oil. Moreover, no significant gender differences were observed for all the investigated pharmacokinetic endpoints [131]. Taken together, these results led to the conclusion that BCP-SEDDS provides a well-tolerated and effective oral delivery system to enhance the oral bioavailability of BCP in humans and may provide a useful tool for maximizing the many health benefits associated with this molecule.

Several BCP-containing formulations have been developed, including Amukkara Choornam [133], CIN-102 [134], and PipeNig^®^-FL [135]. In particular, a 2022 clinical trial involving 30 patients revealed that Amukkara Chooranam, a polyherbal formulation containing BCP, was able to significantly improve clinical assessment parameters in children with autism spectrum disorder [136]. Moreover, PipeNig^®^-FL was shown to stimulate cellular uptake of glucose and to induce membrane translocation of GLUT4 in C2C12 skeletal myotubes, thus preventing lipid accumulation. Due to its high content in BCP, PipeNig^®^-FL represents a very promising bioactive complex that deserves more extensive molecular studies and in vivo investigations in order to support its role as a beneficial metabolic modulator [135].

## 4. Conclusions

The available evidence suggests that BCP can be an important candidate of natural origin endowed with CB2R-selective properties and no psychotropic effects, which may provide a pharmacological rationale for its pharmacotherapeutic application and pharmaceutical development for the treatment of neuropsychiatric conditions. Moreover, its widespread presence in edible plants and safe consumption with no toxicity profile suggest that BCP could be advocated as a nutraceutical and functional food, contributing to overall health and well-being. However, although early animal research indicates that BCP may be effective in treating a large variety of neurological diseases and disorders, there is a need for a more extensive evaluation of the pharmacokinetic profile of BCP and its long-term effects, optimal dosing, alternative routes of administration, and safety in humans, and future trials are required. Furthermore, it has not been fully clarified whether the therapeutic effect of BCP could be due to the improvement of inflammation through modulation of the CB2R or other critical targets of the ECS. Therefore, the specific role of the CB2R in inflammation associated with neuropsychiatric illnesses requires further investigation.

## Figures and Tables

**Figure 1 ijms-25-03203-f001:**
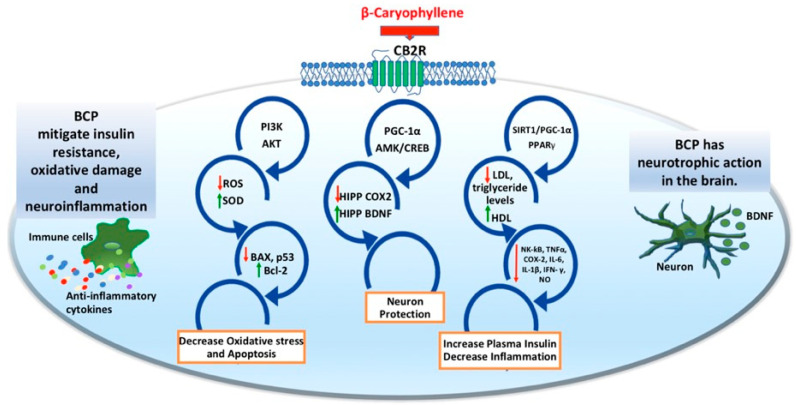
CB2R-mediated neuroprotective and anti-inflammatory mechanisms of BCP. Red and green arrows indicate increased and decreased protein expression levels, respectively.

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
