# Peer review of "Beta-Caryophyllene, a Cannabinoid Receptor Type 2 Selective Agonist, in Emotional and Cognitive Disorders"

_ijms, 2024, doi:10.3390/ijms25063203_

Round 1

Reviewer 1 Report

Comments and Suggestions for Authors

The review titled "Beta-caryophyllene a Cannabinoid Receptor Type 2 Selective Agonist in Emotional and Cognitive Disorders" explores the potential therapeutic applications of Beta-caryophyllene (BCP) in neuropsychiatric disorders, focusing on its role as a CB2R agonist. The review thoroughly discusses the pathophysiology of emotional and cognitive disorders and the neuroinflammatory processes involved, emphasizing the role of the endocannabinoid system, particularly the CB2 receptor. It provides comprehensive insights into the immunomodulatory and anti-inflammatory roles of BCP, supporting its therapeutic potential in managing depression, anxiety, and other psychiatric conditions.

While the review extensively cites various studies, there seems to be a lack of critical analysis of these studies, especially regarding their limitations, sample sizes, and potential biases.

Much of the evidence provided is from animal studies or in vitro models. While this is valuable, the translation of these findings to human clinical contexts is not deeply explored.

The review could benefit from a more detailed discussion on the safety, potential side effects, and long-term implications of using BCP in human populations.

Limitations:

The findings are primarily based on preclinical studies. The applicability of these results to human patients is not fully established.

The review does not address conflicting evidence or alternate viewpoints in current literature, which could provide a more balanced understanding of BCP's potential and limitations.

Comments on the Quality of English Language

No comments

Reviewer 2 Report

Comments and Suggestions for Authors

The review entitled “Beta-caryophyllene, a Cannabinoid Receptor Type 2 Selective Agonist, in Emotional and Cognitive Disorders” provides an extensive review of mostly pre-clinical studies that describes the mechanisms by which BCP promotes anxiolytic and anti-inflammatory activity in a CB2-dependent manner. Before publication, I recommend some major and minor issues should be addressed.

Considering the focus on emotional and cognitive disorders, with special attention to depression and anxiety, I suggest that the authors could improve their definitions for these disorders in the text.

I would also suggest reviewing the initial statements for several paragraphs. There are many examples in which expressions such as “several studies/many studies/significant findings” are followed by only one or two examples to actually support the statement. Thus, it would be advisable to make more restrained allegations.

The section “references” in table 1 should be revised to standardise how they are going to be cited. I could not find any of the references mentioned taking only the number written into consideration.

Major English revision is needed. Authors should reconsider the use of commas before the conjunction “and” throughout the text. It must be done only when it connects two independent clauses. Other language corrections have also been suggested.

Lines 35-37

“In addition, many patients being treatment-resistant contribute to rising costs in national healthcare and economic systems, as they are unable to work and often require hospitalizations.”

Consider rewriting the sentence for better clarity.

Line 39

Consider change the word “causing” for “which caused”.

Line 41

Consider adding the comma after the expression “long-COVID”.

Line 47

Consider removing the comma before the verb “face”.

Line 62

Consider changing the verb “leads” to “lead”.

Lines 65-67

“In recent years, different studies suggested the relationship between alterations in one or more components of the endocannabinoid system (ECS) and some of the symptoms characterize depression and anxiety-related disorders.”

Consider rewriting the sentence for better clarity.

Lines 77-84

The reference mentioned (14) does not account for the localization of CB2R in brain areas cited.

Lines 90-91

Since FAAH has been mentioned as the metabolizing enzyme for anandamide, I suggest adding MAGL as the main catabolizing enzyme for 2-AG.

Lines 103-107

“It has been suggested that CB2R can shift microglia toward an anti-inflammatory phenotype, M2, with a consequent increase in the release of anti-inflammatory cytokines. Both 2-AG and AEA increase the expression of the anti-inflammatory phenotype M2 by microglia, reducing inflammatory microglia markers M1 [16].”

Considering the highly debatable M1/M2 classification for microglial cells, consider removing this nomenclature and keep only the anti-/pro-inflammatory description of phenotypes.

Line 115

Consider adding a comma after “Β-caryophyllene (BCP)”.

Lines 117-119

“Sesquiterpenes are some of the most abundant terpenes in the essential oils of plants, Cannabis included, which in particular consists of about 600 chemical compounds including 140 phytocannabinoids, and more than 100 terpenes.”

Consider rewriting the sentence for better clarity and standardise how the word “Cannabis” is written throughout the text.

Lines 128-129

“the fruits, florescence, seeds, leaves, oil and plant extract rich in BCP”

Consider changing all cited components to plural.

Line 130

Consider correcting “day-to day” to “day-to-day”.

Line 145

Consider removing the comma after “BCP”.

Line 162

Abbreviation for Central Nervous System had already been mentioned, so keep only “CNS”.

Lines 178-179

Write “in vitro” in italic.

Lines 211-225

References skipped from [16] to [34,35].

Lines 215-217

“Depression is a common but serious mood disorder. It causes severe symptoms that affect how a person feels, thinks, and handles daily activities, such as sleeping, eating, or working.”

This sentence is not original; therefore, it should be referenced. (https://www.nimh.nih.gov/health/topics/depression. Accessed in: 17/01/2024)

Line 219

Standardize “CB2R”.

Line 236

This is the first time that THC is mentioned in the text, so it should be written first Δ9-tetrahydrocannabinol.

Line 237

Correct “endocannabinoids” for “phytocannabinoids”.

Lines 236-241

“With respect to THC and cannabidiol (CBD), two of the main endocannabinoids, studies both in human and in rodents demonstrated their antidepressant-like effects. In rats, THC has been shown to have anti-depressant like properties. In humans, reduction of depression was boosted by coadministration of THC and CBD. Finally, CBD has been shown to exert anti-depressant-like effect in several animal models [19].”

Consider rewriting these sentences for conciseness. The first sentence summarizes the following ones.

Line 245

Remove hyphen in “up-regulation”.

Line 253

It is needed to hyphenate “brain derived”.

Line 267

Consider changing the preposition “into” to “of”.

Lines 288-291

“Anxiety includes several other related disorders, such as social anxiety disorder (SAD), phobias and panic, separation anxiety, with post-traumatic stress disorder (PTSD) and obsessive-compulsive disorder being related to but classified separately from anxiety disorders.”

Social anxiety disorder had been already mentioned in the text, therefore only the abbreviation might be used. I would also suggest rewriting the sentence for better clarity, mostly in terms of PTSD and OCD.

Lines 280-282

“If on one hand the administration of CB2R agonists was able to produce an acute anxiolytic effect [58,59], on the other one CB2R antagonists demonstrated to be useful for chronic anxiolytic treatments [54].”

Consider rephrasing this sentence. It leads readers to think that either agonism or antagonism of CB2R induce anxiolytic effect. In fact, the reference used (54) states that AM630 administration increases expression of CB2R and such mechanism could be underlying the anxiolytic response seen in stressed wild type animals. It would also be interesting reading about this dual effect and how it can be explained in terms of pharmacological activity and behavioural outcome.

Line 305

Consider removing the word “effects”.

Line 328

Consider changing “HIV patients” to “patients living with HIV”.

Line 330

Consider changing “between” for “with”.

Lines 334-336

“The immunomodulatory activity of BCP was further explained by an enhanced phagocytic capability, and a conseguent increased lysosomal activity and nitric oxide production in macrophages [68].”

Rephrase the sentence for better clarity. The word “conseguent” seems to be added by mistake.

Line 340

Correct the word “sepsi” to “sepsis”.

Line 371

Consider keeping the abbreviation for “beta-caryophyllene”.

Line 384

Consider keeping only the abbreviation “PD”.

Line 404

Consider change the verb tense from “mitigating” to “mitigate”.

Line 422

The word “cronic” is misspelled.

Line 432

Hyphenate “long term”.

Lines 432-433

“[…] intensifies hippocampal long term depression (LTD), a form of synaptic plasticity associated with cognitive deficits.”

Consider rephrasing this sentence. Even though LTD is associated to cognitive deficits in cases of disruption to brain homeostasis, it is an essential mechanism for memory formation and learning. Therefore, the sentence oversimplifies its biological role.

Line 446

The word “reestabilish” is misspelled.

Lines 448-449/Line 453

The word “rota-road” is misspelled.

Line 452

Write the species “S. odoratissima” in italic.

Lines 465-466

Write the species “Piper nigrum” in italic.

Lines 469-470

“In another study, BCP was found to exhert an anxiolytic effect in zebrafish, significantly decreasing immobility in the novel object recognition (NOR) test [107].”

This sentence does not reflect correctly the findings in this study mentioned. In fact, BCP had little to no effect on anxiety in this animal model, as states the authors in the result section of the cited paper: “βCP had no effect across all variables of interest in the open field test or novel object approach test in any of the treatment groups when compared to the control, aside from a modest decrease in immobility in the novel object approach test in the highest dose used (4.0%).”

Line 476

Reconsider the verb tense of “lead”.

Line 514

Remove hyphen from “in-silico”.

Lines 520-523

“Recently, BCP has been included in the list of compounds regarded as generally recognized as safe by the United States Food and Drug Administration (USFDA), and its safety was further undercored by its approval for use as an additive and preservative in food products and beverages [110].”

Consider rephrasing this sentence for better clarity. I do not recognise the word “undercored”.

Comments on the Quality of English Language

Major English revision is needed. Authors should reconsider the use of commas before the conjunction “and” throughout the text. It must be done only when it connects two independent clauses. Other language corrections have also been suggested in the above Comments box

Round 2

Reviewer 1 Report

Comments and Suggestions for Authors

None.